# Different Characteristics of Annealed Rice Kernels and Flour and Their Effects on the Quality of Rice Noodles

**DOI:** 10.3390/foods12091914

**Published:** 2023-05-07

**Authors:** Ziwen Zhang, Mengshan Shang, Xiaoyu Chen, Lei Dai, Na Ji, Yang Qin, Yanfei Wang, Liu Xiong, Qingjie Sun, Fengwei Xie

**Affiliations:** 1College of Food Science and Engineering, Qingdao Agricultural University, 700 Changcheng Road, Chengyang District, Qingdao 266109, China; 2Qingdao Special Food Research Institute, Qingdao 266109, China; 3State Key Laboratory of Food Science and Technology, Jiangnan University, Wuxi 214122, China; 4School of Engineering, Newcastle University, Newcastle Upon Tyne NE1 7RU, UK

**Keywords:** rice, kernels, flour, annealing, noodles

## Abstract

In this study, the characteristics of indica rice kernels (IRK) and flour (IRF) annealed in different conditions were evaluated, and the quality of rice noodles made with these IRK and IRF was determined. Native IRK and IRF were annealed in deionized water at a kernel or flour to water ratio of 1:3 (*w*/*v*) and temperatures of 50, 55, 60, and 65 °C for 12 and 24 h. Annealing increased the paste viscosity of IRK while decreasing that of IRF. Both annealed IRK and IRF exhibited increases in the gelatinization enthalpy change and relative crystallinity. Annealed IRK gel showed higher hardness, and annealed IRF gel displayed greater springiness. Unlike native rice noodles, annealed IRK noodles exhibited denser pores, while annealed IRF noodles exhibited a looser microstructure. With increasing annealing temperature and time, both annealed IRK and IRF noodles showed enhanced tensile properties. Rice noodles made from IRF annealed at 65 °C for 12 h exhibited a fracture strain of 2.7 times that of native rice noodles. In brief, IRK and IRF exhibited different degrees of susceptibility to annealing. Annealing had more significant effects on IRF than IRK. This study highlights the possibility of using annealed IRK and IRF in rice noodles.

## 1. Introduction 

Rice sustains as a primary source of cereal crops for half of humanity and is consumed in the forms of kernels, flour, and derivatives [1]. Despite being a staple food, indica rice has poor cooking and eating qualities. To enhance cooking convenience and promote the consumption of indica rice, in Asian countries such as China, Vietnam, and Thailand, indica rice is generally used to make rice noodles [2]. Rice noodles, as a starch-based food, provide certain nutritional benefits to consumers, and at the same time, rice proteins that exhibit a well-balanced amino acid profile are largely preserved [3]. At times, coveted rice noodle dishes can be regarded as a tourist attraction to facilitate economic growth. Moreover, the rice noodles market is predicted to increase steadily to USD 3.6 billion in Europe and Asia Pacific [4].

The quality properties of rice noodles are mainly dependent on the textural characteristics of rice flour [5]. However, rice noodles made from native indica rice are less resistant to shear force and exhibit lower tensile strength and extensibility; thus, native indica rice cannot usually meet the requirements for processing [6,7,8]. To obtain high-quality rice noodles, food additives are commonly used, such as modified starches, hydrocolloids, emulsifiers, proteins, and enzymes [9,10,11,12,13,14]. Nonetheless, additives used in foods need to meet strict requirements. Additionally, food additives are not widely accepted by consumers [15]. In contrast, physical modifications such as annealing and heat moisture treatment are widely recognized for their low cost, safety, and effectiveness [16]. Physically modified starches and grain flour are preferred because of the consistency of “clean label” products [17]. For example, Chandla et al. [18] revealed that noodles prepared with heat- and moisture-treated amaranth starch exhibited enhanced hardness and springiness; Purwani et al. [19] found that noodles made with heat- and moisture-treated sago starch showed an increase in elasticity.

As a green physical method to modify starch, annealing (ANN) is a hydrothermal treatment that treats starch with excess water (40–80%) within a scheduled time at appropriate temperatures, which are above the glass transition temperature but below the gelatinization temperature [20]. Annealing facilitates the mutual effects between starch chains within the amorphous and crystalline regions and alters the arrangement of starch molecules [21]. Moreover, annealing could increase the crystallinity without destroying granule structures [22]. Li Wang et al. [23] found that annealing improved rice starch’s heat and shear stability; noodles prepared with annealed rice starch displayed remarkably higher hardness and springiness. Dutta et al. [24] reported that films prepared from annealed potato starches exhibited increased tensile strength and elongation at break.

Although studies have been reported about the annealing of rice starch, annealed rice kernels and rice flour have seldom been studied, especially their differences in properties. Compared with rice kernels, rice flour contains nearly the same components but is the product of a rice kernel with its original structure largely disrupted. We propose that annealing would have different impacts on their physicochemical properties, thereby affecting the quality of rice noodles made of them. Therefore, this study focuses on the physicochemical properties of annealed rice kernels and annealed rice flour and their effects on the quality of rice noodles. The work could be instrumental in establishing an environmentally friendly method to expand the application of annealed rice kernels and rice flour in rice noodles.

## 2. Materials and Methods

### 2.1. Materials

Indica rice was obtained from Guilin Dingyin Food Co., Ltd. (Guilin, China). The native indica rice contained 7.33% protein, 1.00% crude fat, 0.45% ash, 21.76% amylose content, and 74.62% total starch. Unless otherwise stated, deionized water was constantly used during the experiment. All reagents used were of analytical grade.

### 2.2. Preparation of Annealed Rice Kernels and Rice Flour

The annealed indica rice kernels and rice flour were fabricated according to the method of Yadav et al. [25] with some modifications. Briefly, indica rice kernels (50 g) were dispersed in 150 mL of deionized water and then incubated at 50, 55, 60, and 65 °C for 12 or 24 h. To prevent spoilage and acid, deionized water at the same temperature was used to substitute for the water already used every 12 h. Afterward, the annealed rice kernels were quickly cooled in an ice bath, washed with deionized water, and dried in a 45 °C blast drying oven for 6 h.

Indica rice flour was obtained by wet milling of indica rice [26]. Specifically, rice kernels were steeped in water for 2 h and then milled using a mixer grinder (JYZ-D02V, Joyoung Co., Ltd., Jinan, China). Subsequently, the obtained slurry was dried at 45 °C, ground, and passed through a 100-mesh sieve. Indica rice flour was annealed at 50, 55, 60, and 65 °C for 12 or 24 h with constant shaking [24]. After filtration, the annealed rice flour was dried in the same way as used in kernels. Finally, all the prepared samples were ground with a rice mill, passed through a sieve of 100 mesh, and then stored at 4 °C for further analysis. According to the weights before and after annealing, the proportion was estimated. The process yields for the annealed flours obtained from IRK and IRF were about 96.2% and 93.6%, respectively.

### 2.3. Pasting Properties 

To determine the pasting properties, 3 g of the annealed sample powders were mixed with deionized water (25 g) in the test canister using a Rapid Visco Analyzer (Newport Scientific Pvt. Ltd., Warriewood, Australia). The obtained slurry was heated first at a rate of 12 °C/min from 50 °C to 95 °C, held for 2.7 min at 95 °C, and later cooled for 2 min to 50 °C. All the tests were operated at 160 rpm for the rotating speed of the centrifuge. The following pasting parameters were recorded: peak viscosity, trough viscosity, final viscosity, breakdown viscosity, setback viscosity, and pasting temperature [27].

### 2.4. Differential Scanning Calorimetry 

Differential scanning calorimetry (DSC 1; Mettler Toledo, Schwerzenbach, Switzerland) was used to determine the thermal characteristics of the annealed rice kernels and rice flour. The powder samples (3–5 mg) combined with a triploid volume of water were placed in an aluminum pan and then balanced at room temperature overnight before being transferred to a DSC calorimeter. The pan was heated from 25 °C to 125 °C at a heating rate of 10 °C/min. The STARe software was used to calculate the *T*_o_ (onset temperature), *T*_c_ (conclusion temperature), *T*_p_ (peak temperature), and Δ*H* (enthalpy change) [28].

### 2.5. X-ray Diffraction (XRD)

An X-ray diffractometer (AXSD8 ADVANCE, Bruker, Karlsruhe, Germany) was used to analyze the crystalline structures of the annealed rice kernels and rice flour. The powders were placed into a saturated relative humidity chamber for 24 h to achieve a 20% moisture content [29]. The setting parameters are as follows: the scanning range is 4–40°; the acceleration voltage is 40 kV; and the current is 40 mA. 

### 2.6. Fourier Transform Infrared Spectroscopy Analysis (FTIR)

An FTIR spectrometer (NEXUS-870, Thermo Nicolet Corporation, Madison, WI, USA) was used to record the FTIR spectra of the powders between 400 and 4000 cm^−1^. After the annealed kernels and annealed flour were ground into powders, the blending ratio between powders and KBr was generally 0.5–2:100. The powders were first ground in an agate mortar, then KBr was added, ground, and mixed evenly, and later transferred to the tablet press. Each experiment was scanned 32 times with a resolution of 4 cm^−1^ [30]. OMNIC 8.0 software was used to analyze the ordered structure from infrared absorbance values at 1047 and 1022 cm^−1^.

### 2.7. Texture Analysis

After the RVA test, the pastes were stored at 4 °C overnight. The texture analyzer (TAXT Plus, Stable Micro Systems, Surrey, UK) was used to analyze the formed gels. 

The texture profile analysis was performed using a P 0.5 probe with a test speed of 1 mm/s and compression tests under a strain of 50%. The textural parameters of the hydrogels were recorded [31].

### 2.8. Preparation of Fresh Rice Noodles

The fresh rice noodles were processed according to the flat noodle-making method described by Jia et al. [32]. Briefly, 4 g of the powders of annealed rice kernels and rice flour were directly weighed into a beaker and then added to 6 g of deionized water. Silicone molds of length 80 mm × width 18 mm × thickness 2 mm were used to hold the powder slurry, which was steamed in a food steamer for 3.5 min until complete gelatinization. Fresh, wet rice noodles were obtained after cooling to room temperature.

### 2.9. Tensile Properties

The tensile properties of fresh rice noodles were measured using the texture analyzer (TA. XT plus, Stable Micro Systems, Surrey, UK) with the A/TG probe model. The analyzer settings were a probe height of 30 cm, a pre-test speed of 3.0 mm/s, a test speed of 1.0 mm/s, an after-test speed of 3.0 mm/s, and a tensile distance of 30 mm. Tensile strength and elongation at the break of rice noodles were recorded.

### 2.10. Scanning Electron Microscopy (SEM)

The scanning electron microscope (SEM, S-4800, Hitachi Instruments Ltd., Suzhou, China, Science Products Center) was used to observe the morphology of the freeze-dried fresh rice noodles. Fresh wet rice noodles were frozen in liquid nitrogen first, snapped immediately, and then vacuum freeze-dried for 4 d to obtain dried samples. The fractured sections were imaged.

### 2.11. Low-Field Nuclear Magnetic Resonance (LF-NMR)

A low-field nuclear magnetic resonance (LF-NMR) analyzer (Suzhou Niumai Analytical Instruments Co., Ltd.) was used to analyze the transverse relaxation time (T_2_) of the rice noodles. The obtained fresh rice noodles made from annealed kernels and annealed flour with water contents of about 65% and 68%, respectively, were placed into the NMR tube (25 mm diameter) before being transformed into the analyzer. During measurement, the analyzer settings were a maintained temperature of 32 °C and a resonance frequency of 20 MHz. The Carr–Purcell–Meiboom–Gill sequence was used to detect the mobility of water motion in samples [33].

### 2.12. Statistical Analysis

The data obtained were processed using SPSS 16.0 software and then recorded as an average ± standard deviation. Differences were considered statistically significant at 95% (*p* < 0.05).

## 3. Results and Discussion

### 3.1. Pasting Properties of Annealed Rice Kernels and Annealed Rice Flour

The pasting properties of native and annealed rice kernels and flour are shown in Figure 1. Annealed rice kernels at different temperatures and times exhibited higher peak viscosity, final viscosity, and setback but lower breakdown compared with the native rice. At the same temperature except for 60 °C, annealing for 24 h caused a lower final viscosity but a higher breakdown than for 12 h. The results showed enhanced molecular binding forces between starch chains with increasing annealing times [34]. Rice kernels annealed at 60 °C for 24 h showed the lowest breakdown value. This shows that this temperature (60 °C), which is close to but below the onset melting temperature of the native rice, can make a more significant difference to rice kernels.

As for all annealed rice kernels, increased final viscosity and setback could be explained by the re-orientation and re-association of starch chains during annealing [35]. During gelatinization, starch granules swelled upon heating, and the amorphous region absorbed water. As the temperature increased further, swollen starch granules ruptured, leaching out more soluble amylose. Then amylopectin started to swell because too much water entered the tightly bound amorphous regions of double-helical structures, destroying crystalline structures and increasing viscosity [36]. The decreasing breakdown might result from the protein in rice kernels, which restricts the swelling of starch granules. Moreover, decreasing breakdown indicated good heat stability, and increasing final viscosity showed a better gel property. The increasing heat stability reflected the strong interaction between amylose and amylopectin molecules and the integrity of the swollen granules [37].

Annealed rice flour had lower peak viscosity, final viscosity, and breakdown than native rice. As the time and temperature increased, the breakdown of annealed rice flour decreased gradually, and the final viscosity showed an increasing trend. The reduction in the pasting viscosity probably explains the difference in damage extent to the granules [38]. More damaged starch in annealed rice flour reduced the resistance to swelling. Decreasing breakdown indicates hydrogen bonding between water and starch molecules, which strengthened interactions among molecular chains and thus increased heating stability and shearing stability [39].

Annealed rice flour showed different results compared with annealed rice kernels. Annealed rice flour exhibited lower pasting properties than annealed rice kernels, which could be attributed to the interaction between protein and starch in rice, which restricts the swelling of rice starch granules during pasting by further crosslinking, thus reducing the pasting viscosity [38]. During annealing, protein in rice flour was exposed to water and heat more than in rice kernels, and there could be more significant interactions between protein and starch. These changes explained the differences in pasting properties between annealed rice kernels and flour. 

### 3.2. Thermal Properties of Annealed Rice Kernels and Annealed Rice Flour

The gelatinization transition temperatures (*T*_o_, *T*_p_, and *T*_c_), melting range (*T*_c_–*T*_o_), and gelatinization enthalpy (Δ*H*) of rice kernels and rice flour before and after annealing are summarized in Table 1, Table 2, Table 3 and Table 4. In the native starch granules, *T*_o_ represents the heat stability of the crystalline structure. Compared with the native rice, both annealed rice kernels and annealed rice flour exhibited higher *T*_o_, *T*_p_, and *T*_c_ at 60–65 °C. Annealing increased the Δ*H* values of rice kernels and rice flour at 50, 55, 60, and 65 °C for 12 and 24 h. Rice kernels and rice flour remained essentially unchanged in *T*_o_, *T*_p_, and *T*_c_ values after annealing at 50 and 55 °C for 12 and 24 h. In contrast, as the annealing temperature increased from 60 °C to 65 °C, *T*_o_ and *T*_p_ increased apparently. Annealing decreased the melting range (*T*_c_–*T*_o_) from 9.92 to 6.99 and 6.19 for rice kernels and rice flour, respectively, when treated at 65 °C for 24 h, compared to the native rice.

Except for the perfected crystalline area, the amylose–amylose and amylose–amylopectin interactions could also increase *T*_o_, *T*_p_, and *T*_c_ [40]. The higher *T*_o_, *T*_p_, and *T*_c_ values indicate that the crystalline structure of the starch granules suffered less damage, which could be due to the rigid structure of rice grains in the course of processing. Moreover, increased *T*_o_ and *T*_p_ values indicate superior thermal stability. Decreased *T*_p_ could be due to the arrangement of starch granules. Various relative crystallinities and the interactions among amylose-amylose and amylose-lipid in granules could also bring out a decrease in *T*_p_ [41]. In addition, increased Δ*H* indicates that annealed rice kernels and annealed rice flour contained more ordered structures. The different changes in Δ*H* might result from proteins interfering with the bonding of water to starch molecules in protein-starch blends [42]. The more perfect crystalline structure of rice starch resulting from the increasing annealing temperature could cause a decreased melting range [39]. As a result, samples annealed at different conditions showed marked variations, while annealed rice kernels and annealed rice flour exhibited no significant differences in thermal properties.

### 3.3. X-ray Diffraction Analysis of Annealed Rice Kernels and Annealed Rice Flour

The XRD pattern spectra and relative crystallinity values of native and annealed rice kernels and flour are shown in Figure 2 and Table 5 and Table 6. The peaks at 15°, 23°, 17°, and 18.0° (2*θ*) were presented in the native indica rice, indicating the typical A-type crystalline structure. Annealing at 50–65 °C did not alter the crystalline structure, which was consistent with the previous reports [43]. 

A previous report also showed that annealing could hardly change the relative crystallinity of brown rice flour [43]. Annealing increased the relative crystallinity of both rice kernels and rice flour at 50–65 °C for 12 h and 24 h. Increased relative crystallinity was attributed to annealing, which induced the formation of new crystallites and the realignment of the melted crystallites [44]. Enhanced relative crystallinity of starch exhibited a more stable starch crystalline structure, which agreed with the results of increased gelatinization enthalpy. A slight decrease in the rising degree of relative crystallinity after annealing might be owing to the degradation of amylopectin in crystalline regions into amylose in amorphous regions. More amylose tended to entangle the other starch molecules and bind the granules, corresponding to enhanced thermal stability [45]. Moreover, annealed rice kernels had a higher relative crystallinity than annealed rice flour. The difference between annealed rice kernels and rice flour could be starch–lipid complexes, which are not directly proportional to hydrothermal process severity.

### 3.4. Infrared Spectroscopy of Annealed Rice Kernels and Annealed Rice Flour

The ratio of absorbances at 1047/1022 cm^−1^ can be used to quantify the degree of short-range order of starch after annealing [43]. In Figure 3, the infrared spectral curve showed no new peaks. Moreover, annealed rice kernels and annealed rice flour displayed similar FT-IR spectral native samples, demonstrating no chemical bonds formed by annealing [34]. Native and annealed rice kernels and rice flour displayed different ratios of absorbance at 1047/1022 cm^−1^ in Figure 4. Moreover, all annealed samples at 50–65 °C for 12 h and 24 h rendered an increase in the ratio of 1047/1022 cm^−1^. The increased ratio of 1047/1022 cm^−1^ could be due to the more ordered short-range structure and the strengthened internal structure of rice starch. Theoretically, a higher absorbance ratio indicates more significant short-range ordering in starch granules [46]. The increased ratio indicated that the moderate thermal energy and the high moisture content are the dominant factors, causing more efficient packing of double helices within crystalline lamella. The value of the 1047/1022 cm^−1^ ratio is related to relative crystallinity. Annealed rice kernels exhibited a higher 1047/1022 cm^−1^ ratio compared with annealed rice flour and native rice, consistent with the XRD results. 

### 3.5. Texture Analysis of Annealed Rice Kernel Gels and Annealed Rice Flour Gels

The texture characteristics of native and annealed indica rice gels are shown in Table 7, Table 8, Table 9 and Table 10. After annealing, both annealed rice kernel gel and annealed rice flour gel exhibited an increase in hardness, springiness, cohesiveness, gumminess, and resilience. 

The annealed rice flour gel had lower hardness than the annealed rice kernel gel under annealing at 50 °C, 55 °C, and 60 °C for 12 h. When annealed at 50 °C, 55 °C, and 60 °C for 24 h, with increasing annealing time, the annealed rice flour gel had a higher hardness than the annealed rice kernel gel. As the time and temperature increased, the springiness, gumminess, and resilience increased pronouncedly, which plays a crucial role in rice noodles production [47]. Moreover, rice flour gel annealed at 65 °C for 24 h exhibited better springiness than rice kernel gel. Furthermore, annealed rice flour exhibited adequate hardness along with higher springiness and gumminess. The lower hardness of annealed rice flour gel indicates that the protein layer and amylose–lipid complexes around the starch granule surface in rice flour could restrict starch granule swelling during annealing under short processing times and low temperatures [48].

Regarding the higher hardness of annealed rice kernel gel, it is likely that annealing for 24 h could initiate the rearrangement of starch molecules, which may decrease swelling power and solubility, and this decline may promote gel hardness, consistent with the previous study about annealing starch [23]. Increasing the temperature within a certain range promoted the springiness; however, the springiness moderately decreased at 65 °C. Regarding the latter, maybe the heat destroyed the compact structure of protein and starch. The increasing cohesiveness might be due to the formation of hot water-soluble fractions [49]. 

Based on the improved springiness, annealing temperature displayed more significant changes than annealing time in these textural features, indicating that the original binding relationship or molecular size between starch and other non-starch compounds is controlled by thermodynamic effects [50]. Thus, the above results indicate that the difference in starch, protein, and lipid contents between rice kernels and rice flour during annealing highly affected viscosity and textural properties. More protein content in rice kernels is negatively correlated with gumminess, resilience, and other functionalities [51].

### 3.6. Tensile Properties of Annealed Rice Kernel Noodles and Annealed Rice Flour Noodles

The typical stress-strain curves of annealed rice kernel noodles and annealed rice flour noodles are shown in Figure 5. It is obvious that all annealed samples show typical differences in fracture stress and fracture strain.

Native rice noodles have brittle characteristics and poor stretchability. However, in this study, annealed rice kernel noodles and annealed rice flour noodles exhibited good stretchability, which played a significant role in the rice noodles industry [52]. As shown in Figure 5, native rice noodles had an elongation at a break of approximately 40%. As the time and temperature increased, the tensile properties of both annealed rice kernel noodles and annealed rice flour noodles also increased to a certain extent. Annealing increased the fracture stress of rice kernel noodles, which was related to the increased hardness of rice kernel gel. Rice noodles made of annealed rice kernels at 50 °C and 55 °C for 24 h showed excellent stretchability.

Moreover, after rice kernels or flour annealing at 65 °C for 12 h, annealed rice flour noodles showed better fracture strain up to 110% compared with annealed rice kernel noodles, which was consistent with the texture analysis. The increased tensile properties could correspond to the increased relative crystallinity, which meant enhanced integrity of starch granules and a more ordered crystalline starch matrix [53]. Moreover, rice noodles made from rice flour annealed at 65 °C for 12 h showed a low cooking loss. It seemed that higher temperatures promoted the extensibility of rice-flour noodles. Rice kernels were suitable for medium-temperature treatment. The different extensibility could be because annealed rice flour noodles are more pliable and smoother. The annealing for rice flour seemed to fit better for rice noodles with higher stretchability than rice kernels based on the tensile properties [54]. Furthermore, the results indicate that their tensile properties of the rice noodles could be adjusted by controlling the annealing time and temperature for rice kernels and rice flour.

### 3.7. Scanning Electron Microscopy of Annealed Rice Kernel Noodles and Annealed Rice Flour Noodles

The microstructures of the rice noodles cross-sections with different annealing are shown in Figure 6. SEM was used to further investigate the microstructure changes of the transverse section of rice kernel noodles and rice flour noodles in different annealing conditions. In the formation of rice noodles, high temperatures provide a complete gelatinization of starch so that all cooked samples form layered structures and concave surfaces. All the rice noodle samples showed a honeycomb-like structure with large pores/voids in the cross-section of the noodle structure, consistent with previous reports [55]. The SEM results show that annealed rice kernel noodles had much denser microstructures than the rice flour specimens. As the time and temperature increased, the pores in the dried gel increased in quantity and displayed thinner inner walls. The fracture surface of annealed rice flour noodles exhibited a loose microstructure, which might be the swelling of starch granules after a prolonged immersion time [56]. Another reason may be that more protein was affected by water and heat in rice flour, and after that, filtration before drying induced protein loss, allowing starch granules to swell to a greater extent [57].

The gel network of rice noodles could be due to the gelatinization of native rice starch to a large extent, which was similar to those reported by Koh et al. [55], who found that for rice noodles, a continuous and tighter network with an uneven microstructure could be formed. The formation of a starch gel network structure assisted in holding water and maintaining uniform water distribution in the starch gel, preventing the noodle structure from shrinking or cracking during the drying process. After annealing, rice kernels and rice flour both had better morphology (Figure 6), consistent with the results of tensile properties. In conclusion, starch granules and the characteristics of the native rice were significantly altered during annealing, and changes in rice flour differed from those inside the rice kernels.

### 3.8. Water Mobility in Annealed Rice Kernel Noodles and Annealed Rice Flour Noodles

Generally, LF-NMR analysis can be used to study the state, mobility, and distribution of water molecules. The transverse relaxation time (T_2_) distribution profiles could reflect the difference in the degrees of freedom. As shown in Figure 7, the peaks in the horizontal ranges of 0.1–10 ms and 10–100 ms represented the states of the water in hydrogels, with one peak indicating a close relationship between water and starch chains (T_21_), and the other indicating the retention of water by the gel microstructure (T_22_) [58].

After annealing, all rice noodles showed two peaks at T_21_ and T_22_ within the T_2_ relaxation time spectrum. Moreover, the increased peak height at 10–100 ms could be considered as the weakly bound water in the moisture in rice noodles. The denser network in annealed rice absorbed a certain amount of water, increasing water content and mobility. Annealed rice flour showed higher peaks than annealed rice kernels, which might be owing to the interaction between protein and starch, especially amylose molecules. Amylose is known to alter the gel network microstructure and endow a mobility-reducing entity. Annealed rice kernels could retain more amylose, thus restricting the swelling power [59].

The reason for the difference between annealed rice kernels and annealed rice flour could be the inhomogeneous morphological organization. On the one hand, rice kernels may lead to an inhomogeneous diffusion of water and heat during annealing, while rice flour exhibits a little delay in water penetration and air emission [60]. On the other hand, rice flour absorbed more water and then gelatinized under the action of hot steam. The larger holes shown by annealed rice flour noodles under SEM could also be consistent with the mobility and distribution of water molecules.

## 4. Conclusions

In this study, annealing showed a pronounced effect on modifying the characteristics of indica rice kernels and rice flour and the quality of rice noodles. Annealed rice kernels and rice flour showed higher thermal stability, better texture properties, and increased relative crystallinity than native ones. Results also exhibited that rice flour had more susceptibility to modification than rice kernels. Rice noodles made of annealed rice flour at 65 °C for 12 h exhibited better fracture strain. Non-starch components such as proteins and lipids might have an effect on property differences in rice flour during annealing. Therefore, the annealing effects obtained in the present study are helpful in controlling the properties of rice noodles in the food processing industry.

## Figures and Tables

**Figure 1 foods-12-01914-f001:**
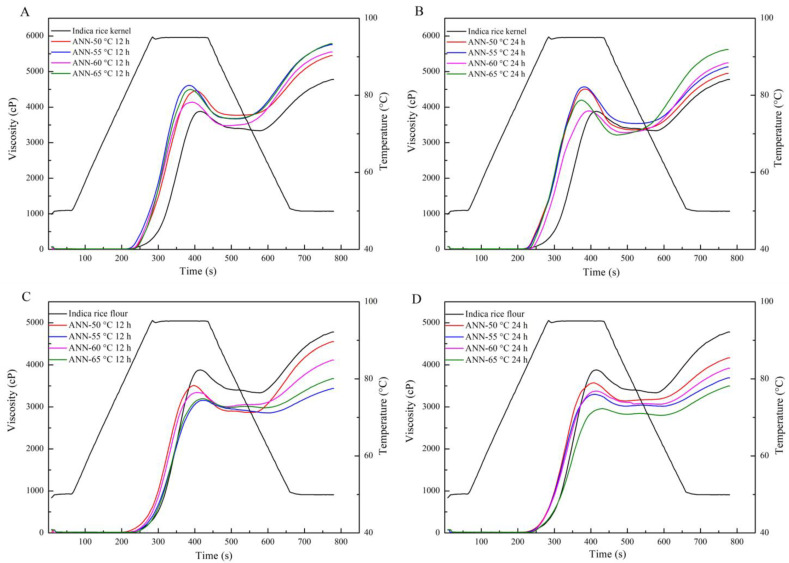
Pasting curves of native and annealed indica rice kernels (**A**,**B**) and annealed indica rice flour (**C**,**D**) (ANN-XY, where X stands for annealing temperature and Y stands for annealing time).

**Figure 2 foods-12-01914-f002:**
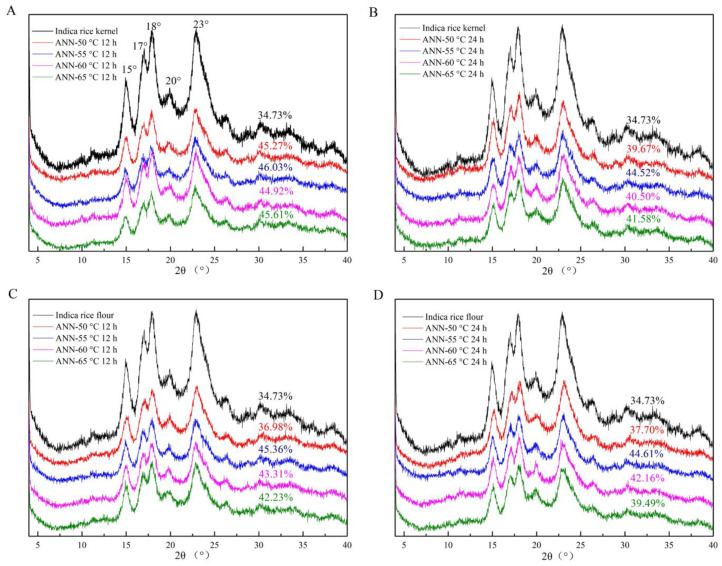
X-ray diffraction patterns of native and annealed indica rice kernels (**A**,**B**) and annealed indica rice flour (**C**,**D**).

**Figure 3 foods-12-01914-f003:**
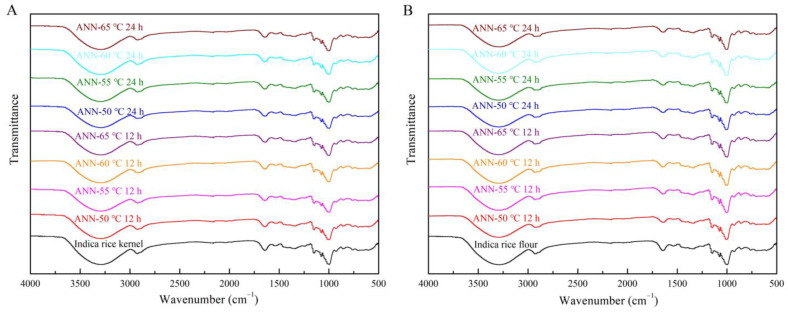
Infrared spectroscopy of native and annealed indica rice kernels (**A**) and annealed indica rice flour (**B**).

**Figure 4 foods-12-01914-f004:**
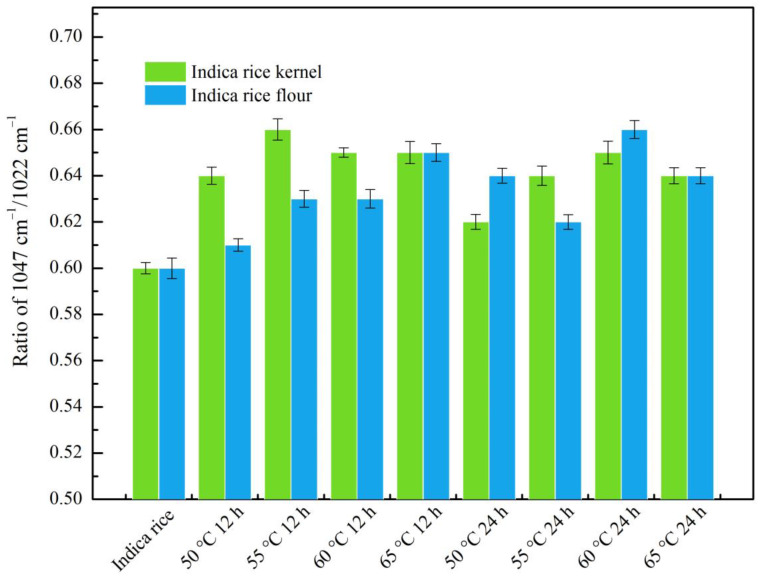
IR intensity ratios at 1047 cm^−1^/1022 cm^−1^ of native and annealed indica rice kernels and annealed indica rice flour. Each value represents the mean of three independent measurements. Bars on top are significantly different (*p* < 0.05).

**Figure 5 foods-12-01914-f005:**
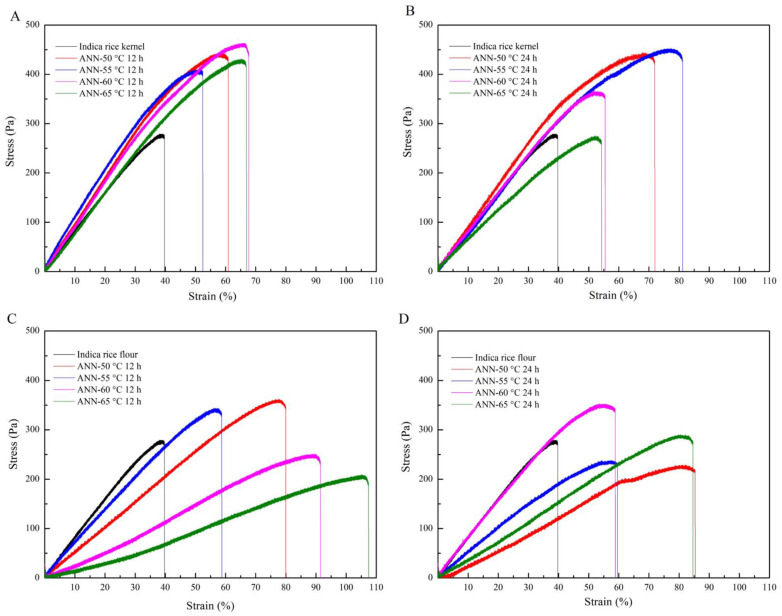
Tensile stress-strain curves of fresh rice noodles made by native and annealed indica rice kernels (**A**,**B**) and annealed indica rice flour (**C**,**D**).

**Figure 6 foods-12-01914-f006:**
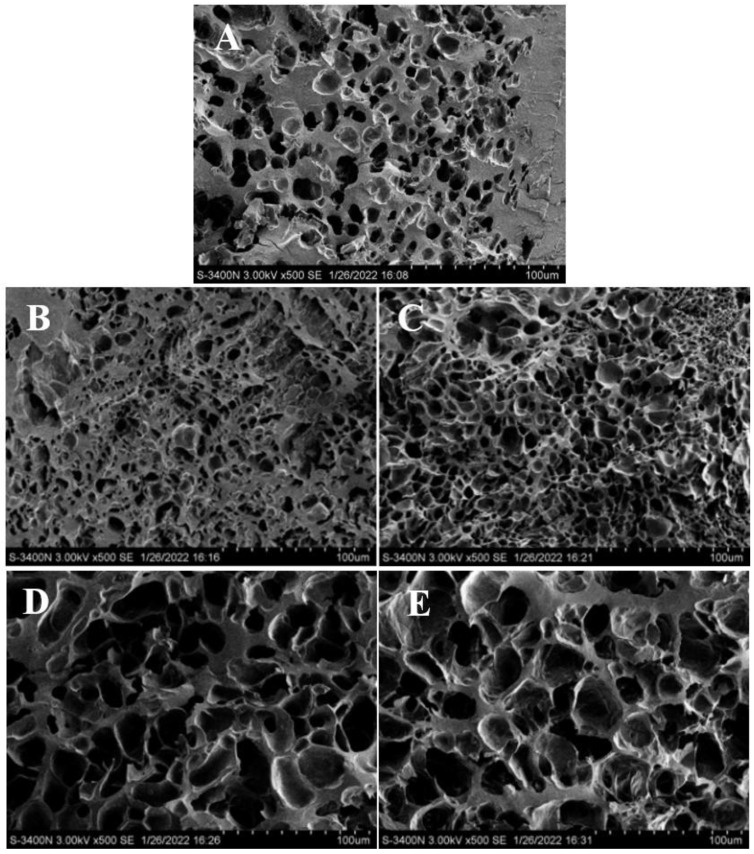
SEM images of native and annealed indica rice kernels and annealed indica rice flour noodles (**A**–**E**) (100 μm). (**A**) Native indica rice; (**B**) ANN-60 °C 12 h indica rice kernels; (**C**) ANN-60 °C 24 h indica rice kernels; (**D**) ANN-65 °C 12 h indica rice flour; (**E**) ANN-65 °C 24 h indica rice flour.

**Figure 7 foods-12-01914-f007:**
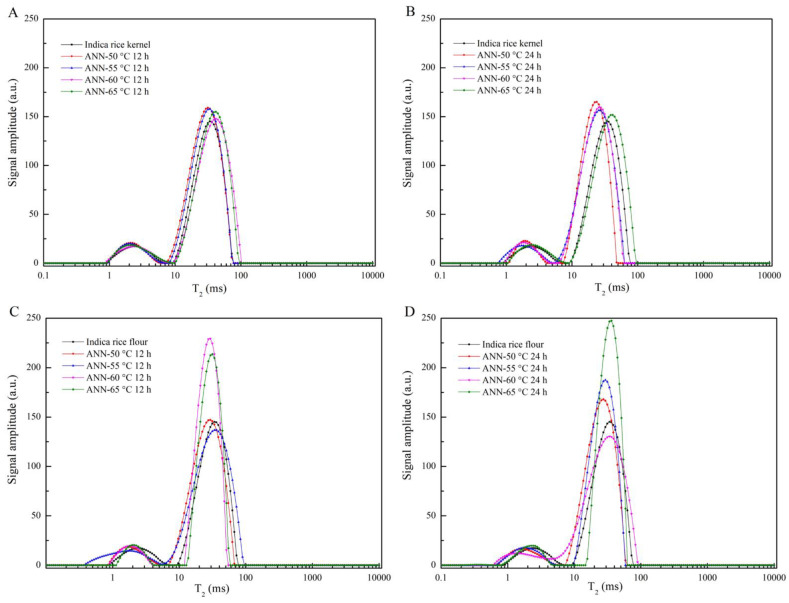
T_2_ results of native and annealed indica rice kernels (**A**,**B**) and annealed indica rice flour (**C**,**D**).

**Table 1 foods-12-01914-t001:** Thermal characteristics of native indica rice and annealed indica rice kernels modified at different temperatures for 12 h (ANN-XY, where X stands for annealing temperature and Y stands for annealing time).

Sample	*T*_o_ (°C)	*T*_p_ (°C)	*T*_c_ (°C)	Δ*H* (J/g)
Indica rice kernel	73.00 ± 0.12 ^a^	77.96 ± 0.11 ^c^	82.92± 0.28 ^ab^	7.69 ± 0.11 ^a^
ANN-50 °C 12 h	73.04 ± 0.14 ^ab^	77.41 ± 0.24 ^ab^	82.24 ± 0.28 ^a^	10.21 ± 0.41 ^b^
ANN-55 °C 12 h	73.33 ± 0.19 ^ab^	77.56 ± 0.21 ^ab^	82.12 ± 0.23 ^a^	10.75 ± 0.11 ^b^
ANN-60 °C 12 h	76.24 ± 0.05 ^c^	79.38 ± 0.09 ^d^	83.35 ± 0.04 ^b^	10.89 ± 0.38 ^b^
ANN-65 °C 12 h	76.71 ± 0.03 ^cd^	79.78 ± 0.10 ^e^	83.43 ± 0.14 ^b^	10.71 ± 1.75 ^b^

The values are represented in the form of the mean ± standard deviation. Values within each column followed by different letters indicate significant differences (*p* < 0.05). *T*_o_—onset temperature; *T*_p_—peak temperature; *T*_c_—conclusion temperature; Δ*H*—enthalpy of gelatinization.

**Table 2 foods-12-01914-t002:** Thermal characteristics of native indica rice and annealed indica rice kernels modified at different temperatures for 24 h (ANN-XY, where X stands for annealing temperature and Y stands for annealing time).

Sample	*T*_o_ (°C)	*T*_p_ (°C)	*T*_c_ (°C)	Δ*H* (J/g)
Indica rice kernel	73.00 ± 0.12 ^a^	77.96 ± 0.11 ^c^	82.92 ± 0.28 ^ab^	7.69 ± 0.11 ^a^
ANN-50 °C 24 h	73.06 ± 0.12 ^ab^	77.25 ± 0.18 ^a^	82.00 ± 0.06 ^a^	11.25 ± 0.08 ^b^
ANN-55 °C 24 h	73.59 ± 0.20 ^b^	77.69 ± 0.03 ^bc^	82.21 ± 0.04 ^a^	10.62 ± 0.04 ^b^
ANN-60 °C 24 h	76.96 ± 0.18 ^c^	79.91 ± 0.23 ^d^	83.44 ± 0.48 ^b^	9.82 ± 0.14 ^ab^
ANN-65 °C 24 h	77.54 ± 0.23 ^d^	80.53 ± 0.18 ^e^	84.50 ± 0.32 ^c^	9.83 ± 0.67 ^ab^

The values are represented in the form of the mean ± standard deviation. Values within each column followed by different letters indicate significant differences (*p* < 0.05). *T*_o_—onset temperature; *T*_p_—peak temperature; *T*_c_—conclusion temperature; Δ*H*—enthalpy of gelatinization.

**Table 3 foods-12-01914-t003:** Thermal characteristics of native indica rice and annealed indica rice flour modified at different temperatures for 12 h (ANN-XY, where X stands for annealing temperatures and Y stands for annealing times).

Sample	*T*_o_ (°C)	*T*_p_ (°C)	*T*_c_ (°C)	Δ*H* (J/g)
Indica rice flour	73.00 ± 0.12 ^b^	77.96 ± 0.11 ^c^	82.92 ± 0.28 ^bc^	7.69 ± 0.11 ^a^
ANN-50 °C 12 h	72.37 ± 0.02 ^a^	76.92 ± 0.18 ^ab^	81.77 ± 0.33 ^a^	10.91 ± 0.11 ^b^
ANN-55 °C 12 h	73.10 ± 0.11 ^bc^	77.24 ± 0.16 ^b^	81.75 ± 0.35 ^a^	11.26 ± 0.36 ^b^
ANN-60 °C 12 h	75.77 ± 0.01 ^c^	78.91 ± 0.08 ^d^	82.46 ± 0.24 ^b^	10.60 ± 0.66 ^b^
ANN-65 °C 12 h	76.82 ± 0.21 ^d^	79.85 ± 0.26 ^e^	83.31 ± 0.44 ^cd^	10.92 ± 0.50 ^b^

The values are represented in the form of the mean ± standard deviation. Values within each column followed by different letters indicate significant differences (*p* < 0.05). *T*_o_—onset temperature; *T*_p_—peak temperature; *T*_c_—conclusion temperature; Δ*H*—enthalpy of gelatinization.

**Table 4 foods-12-01914-t004:** Thermal characteristics of native indica rice and annealed indica rice flour modified at different temperatures for 24 h (ANN-XY, where X stands for annealing temperatures and Y stands for annealing times).

Sample	*T*_o_ (°C)	*T*_p_ (°C)	*T*_c_ (°C)	Δ*H* (J/g)
Indica rice flour	73.00 ± 0.12 ^b^	77.96 ± 0.11 ^c^	82.92 ± 0.28 ^bc^	7.69 ± 0.11 ^a^
ANN-50 °C 24 h	72.46 ± 0.24 ^a^	76.76 ± 0.34 ^a^	81.41 ± 0.49 ^a^	10.80 ± 0.40 ^c^
ANN-55 °C 24 h	73.43 ± 0.04 ^c^	77.23 ± 0.04 ^b^	81.44 ± 0.15 ^a^	11.21 ± 0.82 ^c^
ANN-60 °C 24 h	76.26 ± 0.03 ^d^	79.51 ± 0.08 ^d^	83.13 ± 0.18 ^bcd^	10.83 ± 0.08 ^c^
ANN-65 °C 24 h	77.38 ± 0.33 ^e^	80.25 ± 0.30 ^e^	83.57 ± 0.30 ^d^	9.75 ± 1.00 ^b^

The values are represented in the form of the mean ± standard deviation. Values within each column followed by different letters indicate significant differences (*p* < 0.05). *T*_o_—onset temperature; *T*_p_—peak temperature; *T*_c_—conclusion temperature; Δ*H*—enthalpy of gelatinization.

**Table 5 foods-12-01914-t005:** Relative crystallinity of native and annealed indica rice kernels modified at different temperatures and times.

Sample	Relative Intensity (%)	Sample	Relative Intensity (%)
Indica rice kernel	34.73 ± 0.12 ^a^	Indica rice kernel	34.73 ± 0.12 ^a^
ANN-50 °C 12 h	45.27 ± 0.02 ^b^	ANN-50 °C 24 h	39.67 ± 0.28 ^b^
ANN-55 °C 12 h	46.03 ± 0.11 ^c^	ANN-55 °C 24 h	44.52 ± 0.06 ^d^
ANN-60 °C 12 h	44.92 ± 0.33 ^b^	ANN-60 °C 24 h	40.50 ± 0.04 ^c^
ANN-65 °C 12 h	45.61 ± 0.04 ^b^	ANN-65 °C 24 h	41.58 ± 0.32 ^c^

The values are represented in the form of the mean ± standard deviation. Values within each column followed by different letters indicate significant differences (*p* < 0.05).

**Table 6 foods-12-01914-t006:** Relative crystallinity of native and annealed indica rice flour modified at different temperatures and times.

Sample	Relative Intensity (%)	Sample	Relative Intensity (%)
Indica rice flour	34.73 ± 0.12 ^a^	Indica rice flour	34.73 ± 0.12 ^a^
ANN-50 °C 12 h	36.98 ± 0.34 ^b^	ANN-50 °C 24 h	37.70 ± 0.24 ^b^
ANN-55 °C 12 h	45.36 ± 0.08 ^d^	ANN-55 °C 24 h	44.61 ± 0.21 ^e^
ANN-60 °C 12 h	43.31 ± 0.15 ^c^	ANN-60 °C 24 h	42.16 ± 0.09 ^d^
ANN-65 °C 12 h	42.23 ± 0.11 ^c^	ANN-65 °C 24 h	39.49 ± 0.03 ^c^

The values are represented in the form of the mean ± standard deviation. Values within each column followed by different letters indicate significant differences (*p* < 0.05).

**Table 7 foods-12-01914-t007:** Textural properties of annealed indica rice kernel gel modified by different temperatures for 12 h.

Sample	Hardness (g)	Springiness (g)	Cohesiveness (g)	Chewiness (g)	Resilience (g)
Indica rice kernel	77.06 ± 5.63 ^a^	0.868 ± 0.004 ^a^	0.593 ± 0.033 ^a^	39.60 ± 0.91 ^a^	0.268 ± 0.024 ^a^
ANN-50 °C 12 h	104.25 ± 1.48 ^c^	0.878 ± 0.000 ^ab^	0.597 ± 0.003 ^a^	54.62 ± 0.52 ^c^	0.295 ± 0.008 ^a^
ANN-55 °C 12 h	102.16 ± 2.52 ^c^	0.899 ± 0.013 ^c^	0.619 ± 0.013 ^ab^	56.82 ± 1.68 ^cd^	0.306 ± 0.002 ^ab^
ANN-60 °C 12 h	107.18 ± 0.40 ^d^	0.868 ± 0.011 ^a^	0.594 ± 0.016 ^a^	55.17 ± 0.57 ^c^	0.293 ± 0.006 ^a^
ANN-65 °C 12 h	94.01 ± 4.34 ^b^	0.889 ± 0.000 ^ab^	0.602 ± 0.004 ^a^	50.27 ± 1.94 ^b^	0.280 ± 0.004 ^a^

The values are represented in the form of the mean ± standard deviation. Values within each column followed by different letters indicate significant differences (*p* < 0.05).

**Table 8 foods-12-01914-t008:** Textural properties of annealed indica rice kernel gel modified by different temperatures for 24 h.

Sample	Hardness (g)	Springiness (g)	Cohesiveness (g)	Chewiness (g)	Resilience (g)
Indica rice kernel	77.06 ± 5.63 ^a^	0.868 ± 0.004 ^a^	0.593 ± 0.033 ^a^	39.60 ± 0.91 ^a^	0.268 ± 0.024 ^a^
ANN-50 °C 24 h	103.79 ± 3.29 ^c^	0.888 ± 0.007 ^bc^	0.633 ± 0.008 ^ab^	58.27 ± 2.12 ^c^	0.348 ± 0.042 ^b^
ANN-55 °C 24 h	104.47 ± 0.36 ^c^	0.892 ± 0.007 ^bcd^	0.637 ± 0.013 ^ab^	59.35 ± 1.54 ^d^	0.366 ± 0.004 ^bc^
ANN-60 °C 24 h	85.73 ± 1.90 ^b^	0.909 ± 0.010 ^d^	0.681 ± 0.006 ^b^	53.04 ± 0.15 ^bc^	0.478 ± 0.025 ^d^
ANN-65 °C 24 h	107.84 ± 2.45 ^cd^	0.885 ± 0.002 ^abc^	0.612 ± 0.037 ^ab^	58.34 ± 2.01 ^c^	0.299 ± 0.027 ^a^

The values are represented in the form of the mean ± standard deviation. Values within each column followed by different letters indicate significant differences (*p* < 0.05).

**Table 9 foods-12-01914-t009:** Textural properties of annealed indica rice flour gels modified by different temperatures for 12 h.

Sample	Hardness (g)	Springiness (g)	Cohesiveness (g)	Chewiness (g)	Resilience (g)
Indica rice flour	77.06 ± 5.63 ^a^	0.868 ± 0.004 ^a^	0.593 ± 0.033 ^a^	39.60 ± 0.91 ^a^	0.268 ± 0.023 ^a^
ANN-50 °C 12 h	82.99 ± 1.98 ^b^	0.915 ± 0.002 ^c^	0.644 ± 0.058 ^b^	48.92 ± 5.67 ^b^	0.414 ± 0.116 ^d^
ANN-55 °C 12 h	95.54 ± 5.50 ^c^	0.928 ± 0.008 ^c^	0.626 ± 0.040 ^b^	55.34 ± 0.84 ^bc^	0.316 ± 0.047 ^ab^
ANN-60 °C 12 h	99.60 ± 2.64 ^cd^	0.883 ± 0.015 ^ab^	0.629 ± 0.003 ^b^	55.29 ± 2.10 ^bc^	0.375 ± 0.003 ^c^
ANN-65 °C 12 h	104.62 ± 0.96 ^d^	0.884 ± 0.008 ^ab^	0.612 ± 0.023 ^ab^	56.49 ± 2.13 ^c^	0.304 ± 0.005 ^ab^

The values are represented in the form of the mean ± standard deviation. Values within each column followed by different letters indicate significant differences (*p* < 0.05).

**Table 10 foods-12-01914-t010:** Textural properties of annealed indica rice flour gels modified by different temperatures for 24 h.

Sample	Hardness (g)	Springiness (g)	Cohesiveness (g)	Chewiness (g)	Resilience (g)
Indica rice flour	77.06 ± 5.63 ^a^	0.868 ± 0.004 ^a^	0.593 ± 0.033 ^a^	39.60 ± 0.91 ^a^	0.268 ± 0.023 ^a^
ANN-50 °C 24 h	111.64 ± 1.03 ^c^	0.916 ± 0.001 ^c^	0.688 ± 0.022 ^b^	70.31 ± 3.00 ^d^	0.491 ± 0.032 ^d^
ANN-55 °C 24 h	107.72 ± 4.96 ^bc^	0.891 ± 0.009 ^b^	0.628 ± 0.025 ^b^	60.24 ± 0.28 ^bc^	0.364 ± 0.007 ^c^
ANN-60 °C 24 h	104.65 ± 2.06 ^bc^	0.881 ± 0.003 ^ab^	0.619 ± 0.028 ^a^	57.00 ± 3.90 ^b^	0.304 ± 0.021 ^b^
ANN-65 °C 24 h	103.35 ± 1.06 ^b^	0.937 ± 0.004 ^d^	0.683 ± 0.018 ^b^	66.10 ± 1.36 ^c^	0.346 ± 0.055 ^c^

The values are represented in the form of the mean ± standard deviation. Values within each column followed by different letters indicate significant differences (*p* < 0.05).

## Data Availability

Data is contained within the article.

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
