# Peer review of "Different Characteristics of Annealed Rice Kernels and Flour and Their Effects on the Quality of Rice Noodles"

_foods, 2023, doi:10.3390/foods12091914_

Round 1

Reviewer 1 Report

Starch annealing is well known in the processing of rice-based noodles. However, the changes produced by annealing continue to be studied because of the variety of methods and materials in which they can be useful.

This paper focuses on the differences between grain and flour (obtained by MH) when annealing is applied and its effect on the quality of noodles produced by a traditional Asian method (steaming of flour slurry). The factorial design used to study the process for grain and flour comprised four temperature levels (50, 55, 60, 65°C) and two time levels (12 and 24 h). The authors performed a very complete characterization of the flours (XRD, FTIR, DSC, pasting, gel texture) and rice noodles (extensibility, SEM, NMR).

The scientific quality and data processing are adequate. However, after reading the manuscript carefully, several weaknesses were found, which are listed below. Thus, I cannot recommend this work.

L19 harness or hardness?

L20 IPK or IRK?

L61 Dutta D  or Dutta?

L151 …before being transformed into the analyzer.

What does it mean? Please clarify if fresh noodles were used and report the water content of the noodles.

L159 Before flour characterization, the process yield for the annealed flours obtained from IRK and IRF must be indicated

L173 Please correct …where Y stands for annealing temperature, X stands for annealing time).

L260-262. The difference between annealed rice kernels and rice flour could be starch-lipid complexes, which are not directly proportional to hydrothermal process severity.

Do you have evidence of lipid-amylose complex formation in the systems studied?

To complete the characterization of the optimal noodles, I suggest adding the cooking properties (cooking loss, water absorption) of rice noodles made from rice flour annealed at 65 °C for 12 h.

Reference list: repeated references: 22 - 23, 39 - 40

Reviewer 2 Report

the method of citation not adapted to the requirements of the publisher

line 53 "green method to modify starch," physical I guess?

line 54 "annealing (ANN) is a hydrothermal treatment 53 that treats starch with excess water (>65%) - 40-80% is commonly quoted?

line 69"differences between" why the authors assume differences at the outset, I think that the purpose of the work should focus on getting to know, examining, evaluating and not showing the differences

2.2. Preparation of annealed rice kernels and rice flour - there is no information at what humidity the ANN was conducted at

Line 97-98- "The obtained slurry was heated first at a rate 97 of 12 °C/min from 50 °C to 95 °C, then held for 2.7 min at 95 °C, and later cooled for 2 min 98 to 50 °C." - please explain whether such rapid temperature increases were dictated. these markings are run 1C/ 1 min?... also incredibly fast cooling from 98 to 50 C in 2 minutes? in view of such assumptions, I do not know whether the results will be discussed with any literature

2.6. Infrared spectroscopy analysis with Fourier transform - imprecise determination of the methodology if they were "powders" as the authors indicate or were they pastes? if so, what was their preparation, how much starch material for KBR or maybe it was the FFT-IT ATR technique? then it is possible to directly determine the sample without the need for pelleting.

graph 5 for a and b - should be the same OY scale

Reviewer 3 Report

The manuscript presents the characteristics of indica rice kernels (IRK) and flour (IRF) annealed in different conditions and their effects on the end-use product quality of rice noodles. This work is interesting considering the possibility of using annealed rice kernels and rice flour in industrial production of rice noodles.

The paper is well written, well-structured and the presentation of the results is clear. The language is fluent and precise. The title clearly describes the contents of the paper. The abstract provides a concise and complete summary and the reference list is appropriate. To my opinion, the manuscript fits in thematically with Foods journal. I recommend this paper for publication.

Reviewer 4 Report

The paper's figures and Tables  should be redesigned to be more clear.

The authors should resubmit the paper to be considered for further evaluation.

Table 1 separate data for each time to aid the reader's vsibility

Fig. 1 cannot be read. Suggestion: use four graphs for each temperature and time two for kernel and two for flour

Fig. 2 same as Fig.1 

Crystallinity degree in a separate Table 

Fig. 4 reconsider figure format separate data in groups according to time (all data for the same time together )

Fig. 5 separate 12h from 24 h. then, produce four different figures. Otherwise there are too many curves of different colours. You can used the same scale to show the differences.  Moreover, reduce points' number to droduce the curves. 

-chewiness and gumminess are diffrent attributes the first mostely for solids the second for semi-solid . It is better that the autors choose one of them. 

Round 2

Reviewer 4 Report

I think that the authors did not understand the comments written.
